# Digital nursing and midwifery leadership: Protocol for a multi-method exploration of policy implementation and impact on practice in the WHO European Region

Gillian Janes[1,2°], Lorna Chesterton[1°], Joanne Reid[1,3°] *, Vanessa Heaslip[4°], Michael Shannon[5°], Bente Lüdemann[6°], Rolf-André Oxholm[6°], João Gentil[7°], Clayton Hamilton[8°], Natasha Phillips[4°]

1 School of Nursing, Anglia Ruskin University, Chelmsford, United Kingdom, 2 Centre for Health Systems and Safety Research, Macquarie University, New South Wales, Australia, 3 School of Nursing and Midwifery, Queen's University, Belfast, United Kingdom, 4 School of Health and Society, University of Salford, Greater Manchester, United Kingdom, 5 Faculty of Nursing and Midwifery, Royal College of Surgeons in Ireland, Dublin, Ireland, 6 Norwegian Nurses Organisation, Oslo, Norway, 7 Public Health Unit, ULS Coimbra, Coimbra, Portugal, 8 Regional Office for Europe, World Health Organization, Copenhagen, Denmark

☙ These authors contributed equally to this work
* j.reid@qub.ac.uk

## Abstract

### Background

Digital health is redefining nursing and midwifery practice, fuelled by national and international priorities for health improvement and workforce planning. Developing digitally enabled healthcare systems can help enable universal health coverage and improve safety outcomes while offering solutions to workforce shortages. However, research suggests that nursing/midwifery leaders are often absent from the strategic planning, design, and implementation of digitally enabled healthcare service models and the associated technological systems that directly impact practice.

### Objectives

This paper presents the protocol for a sequential, multi-method exploration of digital health policy implementation and its impact on practice. This investigation from the perspective of national nursing/midwifery leaders, will increase understanding of the impact these professions have on national decision-making, which will be used to inform digital healthcare policy implementation and development across Europe and beyond.

### Methods

A purposive sample of national nursing/midwifery leaders across the WHO European region will be recruited. In Phase 1, individuals will be invited via email to participate

**Data availability statement:** Deidentified research data will be made publicly available when the study is completed and published.

**Funding:** Anglia Ruskin University. Award no HCRC24H5208 The funder had no role in study design, data collection and analysis, decision to publish, or preparation of the manuscript.

**Competing interests:** The authors have declared that no competing interests exist.

in an anonymous online survey, with findings used to inform the topic guide for online focus groups in Phase 2. Descriptive statistical analysis of the survey dataset will be used to understand the range of countries, roles, contexts, participant experiences, and perceptions on which the findings are based. Where possible, analysis will be undertaken, e.g., by country, and participant role to identify any patterns, gaps, and key areas for further exploration during Phase 2. Survey respondents will be offered the opportunity to participate in an online focus group. Free text questions from the survey and data from focus groups will be transcribed verbatim and analysed using a reflexive thematic approach.

## Discussion

The study outlined within this paper will generate empirical data on to what extent and how national nursing/midwifery leaders influence the progress of digital health-care, based on their experiences implementing key European policy. In gaining a better understanding of this policy implementation, and the role played by nursing and midwifery leaders, the factors that facilitate or hinder this process can be identified and better managed, to maximise the benefits of digital healthcare for population outcomes moving forward.

## Ethical approval

Ethical approval for this study was granted on 10.12.25 by Anglia Ruskin University [ID ETH2425−0725]

## Introduction

Digital health has the potential to transform healthcare systems, better meet population health needs, contribute towards workforce planning [1], significantly improve health outcomes, and support the achievement of universal health coverage [2,3]. Indeed, the United Nations General Assembly highlighted the accelerated progress which the technological revolution can afford the health-related Sustainable Development goals [4,5]. The nursing/midwifery profession then, must adapt and keep pace with this technological revolution [6], being at the centre of architecting healthcare systems to ensure they are fit for purpose [7]. As the largest occupational group in healthcare, with a global workforce of 27.9 million, nurses and midwives are pivotally placed to influence and benefit from digitalisation [8]; with digital technologies able to enhance workflow efficiency, evidenced-based treatment decisions, and workforce challenges [2,3,9]. To enable this, nurses and midwives must also be adequately prepared for practice within the digital healthcare arena [10], using continuing education programmes and competency frameworks, which are internationally recognised [11]. Embedding such standards is essential to support decision-making and promote safe, high-quality care [12]. Nursing/midwifery leaders then, play a crucial role in driving national and international strategic digital imperatives [13]

The importance of nursing/midwifery leadership in digital health policy was highlighted by the recent call to action from the International Council of Nurses' (ICN) Digital Transformation Position Statement [1]. This emphasised the lack of nursing/midwifery voice in the WHO Digital Health Action Plan (European Region) 2023–2030 [14] and called for action from National Nursing Associations (NNAs), focusing on four areas: person-centred design, education, inclusivity, and sustainability, using adoption of the International Classification of Nursing Practice (ICNP) as an enabler. Echoing these challenges, Janes et al [15] found major evidence gaps in the literature relating to nursing and midwifery leadership of digital health policy in all four Global Strategic Directions for Nursing and Midwifery [3] domains: leadership, jobs, education, and service delivery.

This protocol is informed by the strategic priorities outlined in the WHO European Region Digital Health Action Plan [14] and the ICN Digital Transformation Position Statement [1], which highlight user-centred design, inclusion, nurse education, and sustainability as key pillars of digital transformation (See Table 2). It builds on the work of international nursing and midwifery leaders who responded to the WHO's recognition of the limited nursing and midwifery voice in shaping the Digital Health Action Plan, an omission that poses risks to the effective and equitable implementation of global digital health policy. By investigating how these core domains are operationalised across national contexts, this study aims to understand the barriers and enablers encountered by national nursing and midwifery leaders, and how they perceive their influence on digital health policy and practice. In doing so, it will contribute empirical evidence to strengthen leadership capacity, improve global health equity, and support the development of an internationally connected and interoperable digital health ecosystem.

The authors acknowledge that midwifery is an independent profession in many countries, with its own regulatory frameworks and professional bodies. However, within WHO European Region member states, midwifery is often not recognised as a distinct profession separate from nursing; instead, the term "nursing" is frequently used in a broad sense to include midwives, especially in national policy and leadership structures. This reality influences how digital health leadership roles are organised and accessed at the national level. In designing this study, we have deliberately included both nursing and midwifery leaders to reflect this regional context. Our recruitment strategy targets national leaders who represent the combined nursing and midwifery workforce to capture insights relevant to both professions as they function in their respective countries. This approach aims to ensure that the perspectives of midwives, whether they are regulated within nursing or as a separate profession, are included and appropriately represented.

## Materials and methods

### Research aims

1. To undertake a sequential, multi-method exploration of the implementation of the ICN Position Statement [1] on Digital Transformation, and the WHO Digital Health Action Plan for the WHO European Region 2023–2030 [14].

2. To inform digital nursing and midwifery healthcare policy implementation and impact across Europe and beyond.

### Operational definition of digital health

*"The field of knowledge and practice associated with the development and use of digital technologies to improve health…Digital health expands the concept of eHealth to include digital consumers, with a wider range of smart and connected devices. It also encompasses other uses of digital technologies for health such as the Internet of Things, advanced computing, big data analytics, artificial intelligence including machine learning, and robotics"*. [2 p11].

## Methods

### Design

The study will comprise two phases, implemented sequentially. Phase 1 will involve an anonymous online survey, with the findings used to inform the topic guide for follow-up online focus groups in Phase 2.

### Recruitment strategy

Participants will be identified through multiple sources across the WHO European Region, including members of the European Federation of National Nursing and Midwifery Associations (EFNNMA), National Nursing Associations (NNAs) affiliated with the International Council of Nurses (ICN), and Chief Nursing and Midwifery Officer (CNO/CMO) networks. National regulatory bodies will be identified via publicly available data on official websites [16]. Personalised email invitations will be sent to potential participants, and recruitment will be further supported through targeted social media posts via the host university and research team networks. The study will aim to include eligible participants from all 53 WHO European Region member states.

It is anticipated that recruitment will commence on February 3rd 2025 and be completed by 28th April 2025, with data collection completed by 30th June 2025, and final results by 30th July 2025.

Participant inclusion/exclusion criteria are shown in Table 1 below.

### Sampling

A purposive sampling approach selects participants according to predetermined criteria and represents a non-probability technique [17]. Adopting this strategy will ensure that participants are deliberately selected because they are 'information-rich' [18 :p 265] and able to offer unique insights into the research topic [19]. The inclusion criteria intentionally do not specify titles such as CNIO or CMIO, as these roles are not universally recognised across the WHO European Region and may inadvertently exclude individuals with strategic national responsibilities. Instead, the study aims to recruit nursing and midwifery leaders operating at the national level who hold a strategic, system-wide perspective on digital health implementation, rather than a frontline clinical role. Selecting participants with an in-depth understanding of the research topic will enable the study to 'yield appropriate and useful information' [20 : p317]. Purposive sampling will also ensure '*better matching of the sample to the aims and objectives of the research, thus improving the rigour of the study and trustworthiness of the data and results.*' [21: p652]. We aim to recruit at least one participant from each of the 53 WHO Regional Office for Europe member states, with a minimum target sample of 25 participants, to support thematic depth and regional diversity. If fewer than 25 participants initially respond, additional targeted invitations will be issued via EFNNMA, ICN, and CNO/CMO networks to secure sufficient numbers. Where more than 25 participants are eligible,

**Table 1. Participant inclusion/exclusion criteria.**

| Inclusion criteria | Exclusion criteria |
|---|---|
| Nursing/midwifery leaders or their representatives working at national level with experience/responsibility for digital health policy.<br>This may include but is not limited to, for example, those in a national professional, regulatory body, or governmental roles. | Nursing/midwifery leaders working in digital health policy but not at national level. |
| Working in a WHO European Region member state | Working in a non WHO Regional Office for Europe member state. |
| English or Russian speaker.<br>Study documents and data collection will be online and available/facilitated in these languages in accordance with WHO Regional Office for Europe protocol. | Participants who are unable to speak in English or Russian |

purposive selection will be applied to ensure inclusion across sub-regions, both nursing and midwifery leaders, and countries with varying levels of digital health maturity. Participants will not be offered any incentives for taking part in the research.

## Study procedure

The online survey will be hosted by Microsoft Forms, a widely used platform considered to be feasible, safe, and convenient and which can be accessed by participants through an online link to the questionnaire or by scanning a QR code [22]. This platform was also considered an appropriate host site due to the flexibility it offers for designing different question types, such as multiple choice, Likert scale and free text [23]. Microsoft Forms is also compatible with different computer device interfaces which is important when working internationally, across numerous countries.

All potential participants will receive information about the study, describing the aims, methods, and rationale, details on confidentiality, and the data handling process. The online survey will be accessed via an email link or QR code and made available in English and Russian, in line with WHO European Region survey protocols. Participants will be informed in the study information sheet that completion and submission of the survey will constitute consent, and a reminder will be provided at the beginning of the survey. The survey will take approximately 15 minutes to complete. Participants will be instructed to avoid identifying any individual, though should this occur, any identifying information will be removed before data analysis to preserve anonymity. The study will be open to responses for 12 weeks, with email reminders sent at 4 weekly intervals.

## Technology equity

To ensure equitable participation across diverse contexts within the WHO European Region, the study will include technology equity considerations throughout recruitment and data collection. Although online focus groups (via Zoom) will be the primary method of engagement (in line with usual WHO Regional Office for Europe practice), participants who face barriers to digital access—such as poor internet connectivity or limited digital literacy—will be offered alternative options, including individual telephone interviews or email-based participation, to accommodate their preferences and needs.

## Data collection

**Phase 1: Online survey.** An online survey will be developed by the research team based on the four domains of the ICN policy statement [1] 1) User-centred design; 2) Nurse education; 3) Inclusivity, and 4) Sustainability; and the four strategic objectives of the WHO Digital Health Action Plan [14] 1) Norms and technical guidance, 2) Country support 3) Networking and knowledge exchange 4) Horizon scanning and scale up. The questions will be designed to explore how and to what extent these key international policy recommendations are being implemented in practice. Table 2 below shows how the ICN Digital Transformation Position Statement [1] aligns with the WHO (2022) Digital Health Action Plan (European Region) 2023–2030 [14].

The survey will consist of open and closed questions to capture examples of good practice, barriers and enablers to implementation, and other mediating factors, including nursing/midwifery leadership of this agenda. Question design will include the use of validated tools such as Likert scales, to measure participants' attitudes and opinions or the extent to which respondents agree or disagree with given statements [24]. The survey will be distributed in accordance with the study recruitment strategy.

A formal translation validation process will be employed, which includes forward translation by bilingual experts familiar with health terminology, review by a second independent translator, and back-translation into English to ensure consistent interpretation between languages. Any discrepancies will be reviewed by the research team in collaboration with regional nursing and midwifery representatives to ensure cultural and contextual accuracy. The translation process will be

**Table 2. Alignment of WHO [14] Digital Health Action Plan (European Region) 2023–2030 and ICN Digital Transformation Position Statement [1].**

| WHO (2022) Digital Health Action Plan (European Region) 2023–2030. | Transformation Position |
|---|---|
| Strategic Priority 1: Setting norms, developing evidence-based technical guidance and formulating direction to support decision-making in digital health | User-centred design<br>Nurse Education<br>Sustainability |
| Strategic Priority 2: Enhancing country capacities to better govern digital transformation in the health sector and advance digital health literacy | Inclusion<br>Nurse Education<br>Sustainability |
| Strategic Priority 3: Building networks and promoting dialogue and knowledge exchange to facilitate interaction between partners, stakeholders and the wider public to steer the agenda for innovation in digital health | Inclusion<br>Nurse Education<br>Sustainability |
| Strategic Priority 4: Conducting horizon-scanning and landscape analysis to identify solutions that are patient-centred and can be scaled up at country or regional level to help shape public health and health systems in the digital era | User-centred design<br>Inclusion<br>Sustainability |

undertaken by qualified bilingual translators from within the WHO European Regional Office, using recommendations by Kunst and Bierwiaczonek [25].

**Phase 2: Online focus groups.** Respondents from Phase 1 will be invited to participate in an online, multi-national, multi-agency focus group of their choice by providing email contact details at the end of the survey. The focus groups are designed to explore key issues, challenges, and enablers associated with the implementation of these digital health policies. The topic guide for this discussion will be based on the findings from the online survey and the focus groups will be facilitated by experienced research staff, to ensure a focused but inclusive and rich discussion: expert facilitation in focus group discussions can deliver a maximum number of perspectives within a limited period [26]. The focus group facilitators will include a English/Russian bilingual speaker where needed, to ensure inclusivity. The focus groups will comprise approximately 6–8 participants and be hosted on the 'Zoom' platform, based on WHO European Region guidance, as this is the most accessible platform across all member states. In addition, if participants are unable to participate in a focus group, individual semi-structured interviews will be offered on the zoom platform.

Focus groups are an appropriate method for this phase to enable an interactive and in-depth exploration of participants' experiences [27]. Participants will have the opportunity to exchange and clarify their ideas within the context of other perspectives, which may not be achieved from individual interviews [27]. Focus groups will also afford participant discussion and identification of improvements for policy and practice within an international context to support the development of widely applicable study outcomes. As Cameron [28] observes the interaction of participants as peers stimulates discussion to produce new ideas and insights. Additionally, using online focus groups will facilitate participation by individuals located in disparate geographical locations and time zones, by offering a flexible and practical means of in-depth data collection [29]. To protect anonymity, participants will be asked to disable their cameras and remove any identifying information, such as their name or role, from their Zoom profile (in the study PIS and before the focus groups start). Insights from each focus group will be used to shape and refine subsequent discussions to deepen the richness of the data collected.

To address power dynamics within and across focus groups, this study will adopt strategies to create a respectful and inclusive environment [30]. Skilled facilitators will establish clear ground rules, use inclusive techniques such as turn-taking, prompts, and will actively encourage contributions from quieter participants. They will also monitor group dynamics to minimise dominance and promote equitable participation. These approaches will foster open dialogue and support the

collection of authentic perspectives from senior nursing and midwifery leaders, aligning with established best practices in qualitative research.

## Data analysis

Microsoft Excel and Microsoft Forms software will be used to collect data and support the analysis of the online survey. Descriptive statistics will be completed across the survey dataset and used to understand the range of countries, roles, contexts, and participant experiences and perceptions on which the findings are based. Where possible, sub-group analysis will be undertaken, e.g., by country, and participant role, to enable any patterns, gaps, and key areas for further exploration during Phase 2 to be identified. Free text responses to open questions will be analysed using reflexive thematic analysis [31] with NVivo software used to support this.

Focus groups will be audio recorded and transcribed verbatim by the study team supplemented by the 'Zoom auto-transcription service'. Summary notes taken by the focus group facilitators will be fed back to participants at appropriate points during the discussion, to enable in-group member-checking and verification by participants. Anonymised transcripts will be uploaded into NVivo software and analysed using Braun and Clarke's [31] 6-stage reflexive method for identifying, analysing, and reporting patterns and themes from the data (see Table 3). Identified themes will be reviewed against anonymised transcripts and a final theme structure agreed amongst the research team. Any disagreements regarding themes and subthemes will be discussed until consensus is achieved. Table 3 describes the planned process of analysis.

## Rigour

The rigour of the study will be ensured through strict monitoring of the data collection and analysis process and adherence to the study protocol. All focus groups will be audio recorded and transcribed verbatim, supported by the zoom auto transcription service, and manually checked for accuracy. Field notes will be taken throughout each focus group, which will add observational perspectives and context to the transcripts to ensure credibility [32]. McMullin [33] asserts the importance of scholars documenting the transcription process, to recognise the ethical considerations of the process and the need for transparency. During the focus group discussion, facilitators will clarify and confirm with participants, the intended meaning of opinions expressed, by using reflective questioning techniques [34].

Table 3. Application of Braun and Clarke's (2006) six-phase approach to thematic analysis.

| Stage 1: Familiarisation with the data | All focus groups will be audio-recorded and transcribed verbatim. All transcripts will be checked for accuracy. Researchers (LC, NP, GJ, VH) will familiarise themselves with the data. |
| --- | --- |
| Stage 2: Initial coding of the data | Researcher (LC) will create initial coding of the data using NVIVO software. Researchers (GJ, NP, VH) will independently check and verify coded data, with input from the wider research team. |
| Stage 3: Data searched for themes | Data will be searched for initial themes by LC and then discussed with additional researchers (GJ, JR, R-A O). Initial themes will then be reviewed for coherence and credibility by the wider team. |
| Stage 4: Themes reviewed and refined | Themes will be reviewed by the research team and refined/organised into an overall theme structure to address the research questions, led by JR. |
| Stage 5: Themes defined and named | Final themes will be defined based on the consensus of the research team. |
| Stage 6: Academic article produced | When data analysis has been completed, an initial draft paper for publication will be produced (GJ, LC, VH, NP). The remaining team members (CH, R-A O, BL, MS, JR, JG) will contribute to subsequent refinement and final editing. |

The analysis and interpretation of qualitative data, collected through open questions in the online survey (Phase1) and the focus group discussion (phase 2) will be independently coded by researchers (detailed in Table 3). Using Braun and Clarke's [31] framework will allow a reflexive approach to guide thematic analysis, strengthening rigor and reducing potential bias. Fischer [35] describes reflexivity as the continual reflection of researchers on their engagement with the process of data collection and analysis. Whilst, Probst and Berenson [36] add that reflexivity can serve as a process to establish trustworthiness and rigour. The final aspect of ensuring the rigour of the study, and underscoring the credibility of the reported evidence, will be in the use of anonymised direct quotations from participants when reporting the study.

## Replicability

To support replicability, the full study protocol and data collection tools will be made publicly available through open-access publication and institutional repositories. This will include a transparent description of participant recruitment, data management procedures, and coding frameworks. Data analysis will be conducted using systematic methods that are clearly documented, allowing future researchers to replicate or adapt the study design in different contexts [37] Additionally, comprehensive records of coding decisions and reflective notes will be maintained as part of the audit trail, ensuring transparency and methodological consistency [38]

## Ethical considerations

The study has received ethical approval from [Anonymised} Ethics ID ETH2425−0725. All aspects of good practice in research will be followed at all times, in line with the research integrity guidelines of the host University [Anonymised reference]. Participants taking part in the survey and focus groups, will be informed that any information given will be collected anonymously where possible or pseudo-anonymised, and treated confidentially. Respondents completing the online survey will be asked to click on a link to take them outside the anonymous survey where they can provide their contact details if they wish to volunteer for a focus group. This strategy will preserve survey anonymity. All focus groups will be audio-recorded on the 'Zoom' online platform used to host the sessions. All audio recordings will be transcribed verbatim by the research team, supplemented by the Zoom transcription function. Transcripts will be verified by another member of the research team, to confirm accuracy. All data collected as part of this study will be subject to the UK Data Protection Act [39] and handled following UK General Data Protection Regulations policy [40]. We intend to disseminate the findings of this research in conference presentations, webinars, and peer-reviewed journals, which may require the use of direct quotations from participants. In this case, a pseudonym will be ascribed to each quotation, to maintain anonymity. All data will be subject to the host university privacy policy.

## Discussion

Digital health technologies and new digitally enabled service delivery models have gained increasing prevalence as a means of improving the quality, efficiency and safety of healthcare provision [41]. To realise these advantages, nurses/midwives need to be digitally competent and proficient in informatics and technology to enable critical decision-making [42]. As Heaslip and colleagues [43] observe, to enable nurses and midwives to optimise their scope of professional practice, the profession must contribute to and benefit from digital technology, which supports rather than impedes the nursing/midwifery process. Nursing/midwifery leaders then, need to be fully engaged in the co-design, planning, deployment, and evaluation of these digital health technologies [44]. However, there is scant evidence about the role and impact of nurse leadership in digital health development [45] and such omissions compound the invisibility of the contributions that nursing/midwifery leaders are making to strategic digital health policy and practice development [15,45].

From a global perspective, there are marked disparities in the degree of digital health progress, with many countries still requiring 'institutional support' for the implementation of their national strategies [2 p10]. Booth et al [46] note the

complex challenges facing regions and countries, including internet connectivity, interoperability, as well as health information and digitalisation processes. This situation highlights the need to develop global standards, which can be implemented by nursing/midwifery leaders across the diverse levels of digital maturation [15]. By identifying specific factors that facilitate or impede the implementation of key digital health policies internationally, the findings from this study can be used to create shared learning and support for best practices across the WHO European Region member states and potentially beyond. This reflects the perspective of the ICN [1] whereby nursing/midwifery leadership is seen as the crucial element in ensuring that the nursing/midwifery profession makes an active contribution to the digital transformation of healthcare and workforce development for the benefit of service users. It is therefore critical that the contextualised experiences of these key leaders are examined to inform wider national and international strategies to support their leadership of policy and practice.

## Conclusion

Using the two main international digital health policy directives, this study will focus on the key areas outlined by the ICN call for action in their digital transformation position statement [1] and the key priorities of the WHO Digital Health Action Plan [14]. Better understanding the experiences and perceptions of national nurse and midwifery leaders, will enable a better understanding of how digital healthcare is progressing across the WHO European region, and how much influence they are having on policy development and strategic implementation, offering an evidence-based, contextual evaluation of current progress regarding the implementation of key digital health policy.

## Acknowledgments

Our sincere thanks go to: The European Federation of National Nursing and Midwifery Associations (EFNNMA) and National Nursing Association (NNA) members of the International Council of Nurses (ICN, 2024) from within the WHO European Region, Ms Maggie Langins, Nursing and Midwifery Policy Advisor, WHO European Region, Division of Country Health Policies and Systems, WHO European Regional Office; Health and Care Research Centre, Faculty of Health, Medicine and Social Care, Anglia Ruskin University for supporting this study.

## Author contributions

**Conceptualization:** Gillian Janes, Joanne Reid, Vanessa Heaslip, Michael Shannon, Bente Lüdemann, Rolf-Andrè Oxholm, João Gentil, Natasha Phillips.

**Funding acquisition:** Gillian Janes.

**Methodology:** Gillian Janes, Lorna Chesterton, Joanne Reid, Vanessa Heaslip, Michael Shannon, Bente Lüdemann, Clayton Hamilton, Natasha Phillips.

**Supervision:** Gillian Janes, Joanne Reid.

**Writing – original draft:** Gillian Janes, Lorna Chesterton, Joanne Reid, Vanessa Heaslip, Michael Shannon, Bente Lüdemann, Rolf-Andrè Oxholm, João Gentil, Clayton Hamilton, Natasha Phillips.

**Writing – review & editing:** Gillian Janes, Lorna Chesterton, Joanne Reid, Vanessa Heaslip, Michael Shannon, Bente Lüdemann, Rolf-Andrè Oxholm, João Gentil, Clayton Hamilton, Natasha Phillips.

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
