## [Decision Letter · Decision Letter 0]

24 Jul 2025

Dear Dr. Reid,

The reviewers raised several major issues, detailed below. 

We look forward to receiving your revised manuscript.

Kind regards,

Moustaq Karim Khan Rony, RN, MSS, MPH

Academic Editor

PLOS ONE

Journal Requirements:

“Anglia Ruskin University. Award no HCRC24H5208”

Reviewers' comments:

Reviewer's Responses to Questions

**Comments to the Author**

1. Does the manuscript provide a valid rationale for the proposed study, with clearly identified and justified research questions?

Reviewer #1: Yes

Reviewer #2: Partly

Reviewer #3: Yes

2. Is the protocol technically sound and planned in a manner that will lead to a meaningful outcome and allow testing the stated hypotheses?

Reviewer #1: Yes

Reviewer #2: Partly

Reviewer #3: No

3. Is the methodology feasible and described in sufficient detail to allow the work to be replicable?

Reviewer #1: Yes

Reviewer #2: No

Reviewer #3: Yes

4. Have the authors described where all data underlying the findings will be made available when the study is complete?

Reviewer #1: Yes

Reviewer #2: Yes

Reviewer #3: Yes

5. Is the manuscript presented in an intelligible fashion and written in standard English?

Reviewer #1: Yes

Reviewer #2: Yes

Reviewer #3: Yes

You may also provide optional suggestions and comments to authors that they might find helpful in planning their study.

Reviewer #1: 1. The plan of how to recruit the needed sample of using leaders is not clear.

2. It would be better to conduct the key informant interview along with focus group.

Reviewer #2: This protocol addresses a highly relevant policy gap in digital nursing/midwifery leadership, with ‎strong alignment to WHO/ICN priorities. However, essential revisions are required to ensure ‎methodological rigor and replicability:‎

‎(1)‎ Introduction (p. 1): Frame explicit research questions aligned with the priorities and domains ‎in Table 2 (e.g., What barriers do national nursing/midwifery leaders face in implementing X ‎and Y?).‎

‎(2)‎ Sampling criteria (Table 1): Clarify the inclusion criteria for national leaders (e.g., years of ‎experience, job titles) and replace ambiguous terms like "designates" with specific role ‎definitions. While this sampling strategy aligns with the goal of evaluating deep policy ‎insights, it may miss frontline implementation barriers. Including focus groups with frontline ‎nurses could help triangulate "impact on practice" claims—otherwise, provide a clear ‎justification for their exclusion.‎

‎(3)‎ Methods section: Attach the survey tool as supplementary material (important for replicability). ‎Although the purposive sampling strategy appropriately targets "information-rich" national ‎leaders, the protocol must define minimum sample ranges to ensure thematic depth and ‎cross-country representativeness. Refer to similar WHO policy studies for guidance.‎

‎(4)‎ Consider adding inferential statistics for regional comparisons (e.g., Chi-square tests).‎

‎(5)‎ Study Procedures (p. 4): Address tech equity considerations—such as offering offline ‎alternatives (e.g., phone interviews)—and detail the translation validation process. Also, ‎provide an explicit strategy to mitigate power dynamics within and across focus groups.‎

Reviewer #3: Title: Digital nursing and midwifery leadership: Protocol for a multi-method exploration of policy implementation and impact on practice in the WHO European Region

This is an interesting and timely topic that needs attention.

Nursing and Midwifery professions

• The protocol refers to nursing/midwifery throughout, which seems to infer that midwifery is subsumed into nursing, which is not accurate. The only international professional organisation that is mentioned is the International Council of Nurses (ICN). There are many midwives who are not nurses, and many nurses who are not midwives. It would be preferable to have the inputs from members and leaders of each of the two professions in order to capture the data and issues as fully as possible, and to reflect accurately what the title of the protocol claims. If this protocol is earnest about researching this topic for midwifery and midwives, it should consult the International Confederation of Midwives (ICM) policy documents regarding digital capacity and essential competences for practice, and include midwives in the focus group discussions.

• This is a protocol which projects the following timeline: “Recruitment has not been completed. It is anticipated that recruitment will be completed by April 2025, with data collection completed by June 2025, and final results by July 2025”. The protocol was sent for publication review in July 2025. Given the omission of midwifery associations and the International Confederation of Midwives, the protocol has set up an avoidable limitation / weakness, which is unfortunate. This has implications for sampling, the development of the online survey tool, and analysis.

Language

• Languages are limited to English and Russian (“in line with WHO European Region survey protocols”). This excludes the participation of people who might be information-rich in terms of the topic, but who are not fluent in either of these two languages. Given that WHO has six official languages (Arabic, Chinese, English, French, Russian, and Spanish) https://www.who.int/about/communications/understandable/audiences-language it is puzzling that two European languages (French and Spanish) are not accommodated in this protocol.

• Language is not listed as an inclusion or exclusion factor. This should be explicit.

• In the focus group discussions (FGD), it is noted that a bilingual facilitator will be required if necessary. However translation facilities will be required for participants so that they can properly engage in a focus group discussion, otherwise they will effectively be excluded even if they are part of the FFD.

Focus group discussions

• The strategy of summary notes taken by the focus group facilitators being fed back to participants at appropriate points during the discussion, to enable in-group member-checking and verification by participants is welcomed.

• Will feedback be done across groups to enrich subsequent FGDs?

Privacy

While the use of Zoom for FGD supported, one needs to make technical arrangements to ensure that participants’ names are not reflected on screen as part of the video-record. Please ensure that a separate research identifier is used during data collection (FGD) so that the inputs can be ascribed to a participant’s research identifier. This requires explanation.

**Do you want your identity to be public for this peer review?** For information about this choice, including consent withdrawal, please see our Privacy Policy

Reviewer #1: No

Reviewer #2: **Yes: ** Nada Alaidarous

Reviewer #3: No

---

## [Author Response · Author response to Decision Letter 1]

13 Aug 2025

Feedback/recommendation Response

Journal requirements

“Anglia Ruskin University. Award no HCRC24H5208”

Please include this amended Role of Funder statement in your cover letter; we will change the online submission form on your behalf. Thank you for drawing our attention to this. We have added the words: The funders had no role in study design, data collection and analysis, decision to publish, or preparation of the manuscript

P7

This information has been added to the cover letter

Ethical considerations appears within the methods section, so no action taken regarding its position in the manuscript.

Information added to manuscript page 10

5. If the reviewer comments include a recommendation to cite specific previously published works, please review and evaluate these publications to determine whether they are relevant and should be cited. There is no requirement to cite these works unless the editor has indicated otherwise. No action required

Reviewer #1

1. The plan of how to recruit the needed sample of using leaders is not clear.

Thank you for your comment. To add greater clarity we have re-written the recruitment strategy section page 3

2. It would be better to conduct the key informant interview along with focus group.

This is a useful observation, and text added to section headed: phase 2 Online focus groups, p5 . This now includes 1:1 interviews which will be offered to volunteers who are unable to participate in a focus group. A rationale for adding this alternative method of data collection has been added as well.

Reviewer #2: This protocol addresses a highly relevant policy gap in digital nursing/midwifery leadership, with ‎strong alignment to WHO/ICN priorities. However, essential revisions are required to ensure ‎methodological rigor and replicability:‎

Thank you for this positive feedback. We have added some text to complement the existing section on rigour on page 6 to discuss replicability of the study When we publish the completed study we will add the online survey as supplementary data, which will again increase the replicability of the study.

1.‎ Introduction (p. 1): Frame explicit research questions aligned with the priorities and domains ‎in Table 2 (e.g., What barriers do national nursing/midwifery leaders face in implementing X ‎and Y?).‎

Thank you for this insightful comment. To address this we have reframed the last paragraph of the introduction to more explicitly show how the research aligns with the strategic priorities in key policy page 2

‎2.‎ Sampling criteria (Table 1): Clarify the inclusion criteria for national leaders (e.g., years of ‎experience, job titles) and replace ambiguous terms like "designates" with specific role ‎definitions. While this sampling strategy aligns with the goal of evaluating deep policy ‎insights, it may miss frontline implementation barriers. Including focus groups with frontline ‎nurses could help triangulate "impact on practice" claims—otherwise, provide a clear ‎justification for their exclusion.‎

We have clarified our sampling methodology to remove any ambiguity. The sampling strategy deliberately does not specify role or length of service as this would inadvertently exclude participants, in the absence of a universally recognised title. Your comments have strengthened this section, and we are appreciative of your input.

‎3.‎ Methods section: Attach the survey tool as supplementary material (important for replicability). ‎Although the purposive sampling strategy appropriately targets "information-rich" national ‎leaders, the protocol must define minimum sample ranges to ensure thematic depth and ‎cross-country representativeness. Refer to similar WHO policy studies for guidance.‎

Your observations are useful, and we will publish the online survey tool to improve replicability. We have added text to the sampling section and specified a minimum target to ensure thematic depth and regional diversity.

‎4.‎ Consider adding inferential statistics for regional comparisons (e.g., Chi-square tests).‎

Thank you for this interesting point. Just to clarify, this was an exploratory not a comparative study. We were not attempting regional comparisons and the small sample size/number of responses meant this this not appropriate.

‎5. Study Procedures (p. 4): Address tech equity considerations—such as offering offline ‎alternatives (e.g., phone interviews)—and detail the translation validation process. Also, ‎provide an explicit strategy to mitigate power dynamics within and across focus groups.‎

This is a good point. To reflect this view, we have added a new section entitled: Technology Equity page 5

Strategies to mitigate power dynamics added to focus group section on page 7

Reviewer #3: This is an interesting and timely topic that needs attention.

Thank you for this positive feedback.

Nursing and Midwifery professions

• The protocol refers to nursing/midwifery throughout, which seems to infer that midwifery is subsumed into nursing, which is not accurate. The only international professional organisation that is mentioned is the International Council of Nurses (ICN). There are many midwives who are not nurses, and many nurses who are not midwives. It would be preferable to have the inputs from members and leaders of each of the two professions in order to capture the data and issues as fully as possible, and to reflect accurately what the title of the protocol claims. If this protocol is earnest about researching this topic for midwifery and midwives, it should consult the International Confederation of Midwives (ICM) policy documents regarding digital capacity and essential competences for practice, and include midwives in the focus group discussions.

Thank you for highlighting the distinction between nursing and midwifery as separate professions. We acknowledge that midwifery is an independent profession in many countries, with its own regulatory frameworks and professional bodies. However, within the WHO European Region, midwifery is often not recognised as a distinct profession separate from nursing; instead, the term “nursing” is frequently used in a broad sense to include midwives, especially in national policy and leadership structures. This reality influences how digital health leadership roles are organised and accessed at the national level, and we do state that we also consulted the Chief nursing and midwifery leadership networks, and European Federation of National Nursing and Midwifery Associations (EFNNMA)across Europe. We have added more text to clarify this issue.

• This is a protocol which projects the following timeline: “Recruitment has not been completed. It is anticipated that recruitment will be completed by April 2025, with data collection completed by June 2025, and final results by July 2025”. The protocol was sent for publication review in July 2025. Given the omission of midwifery associations and the International Confederation of Midwives, the protocol has set up an avoidable limitation / weakness, which is unfortunate. This has implications for sampling, the development of the online survey tool, and analysis.

The manuscript was submitted on 30/01/2025 and prior to the study commencing. However, we have no influence over the journal’s review/publication timeline. The study funding was time limited, hence the timeline indicated in the manuscript.

Language

• Languages are limited to English and Russian (“in line with WHO European Region survey protocols”). This excludes the participation of people who might be information-rich in terms of the topic, but who are not fluent in either of these two languages. Given that WHO has six official languages (Arabic, Chinese, English, French, Russian, and Spanish) https://www.who.int/about/communications/understandable/audiences-language it is puzzling that two European languages (French and Spanish) are not accommodated in this protocol.

The study has been completed with the support of the WHO policy office, which have given general guidance as to how their surveys and focus groups are routinely administered. We acknowledge that specific projects may include greater inclusivity regarding language translation. Please note that no funding has been received from the WHO to complete this study

• Language is not listed as an inclusion or exclusion factor. This should be explicit. Table 1 displaying inclusion/exclusion criteria changed to reflect this point.

• In the focus group discussions (FGD), it is noted that a bilingual facilitator will be required if necessary. However translation facilities will be required for participants so that they can properly engage in a focus group discussion, otherwise they will effectively be excluded even if they are part of the FFD

In the section headed ‘focus groups’, there is information on bilingual facilitators, and the use of expert translators has been strengthened on page 6

Focus group discussions

• The strategy of summary notes taken by the focus group facilitators being fed back to participants at appropriate points during the discussion, to enable in-group member-checking and verification by participants is welcomed.

Thank you for this positive feedback.

• Will feedback be done across groups to enrich subsequent FGDs?

Yes this is a good point, and we have added that insights from each focus group will shape and refine subsequent discussions to deepen the richness of the data collected page 7

Privacy

While the use of Zoom for FGD supported, one needs to make technical arrangements to ensure that participants’ names are not reflected on screen as part of the video-record. Please ensure that a separate research identifier is used during data collection (FGD) so that the inputs can be ascribed to a participant’s research identifier. This requires explanation.

To protect anonymity, participants will be asked to disable their cameras and remove any identifying information, such as their name or role, from their Zoom profile. Text added to page 7

---

## [Decision Letter · Decision Letter 1]

31 Aug 2025

Dear Dr. Reid,

Thank you for submitting your manuscript to PLOS ONE. After careful consideration, we feel that it has merit but does not fully meet PLOS ONE’s publication criteria as it currently stands. Therefore, we invite you to submit a revised version of the manuscript that addresses the points raised during the review process.

We look forward to receiving your revised manuscript.

Kind regards,

Oluchukwu Loveth Obiora, PhD

Academic Editor

PLOS ONE

Journal Requirements:

Reviewers' comments:

Reviewer's Responses to Questions

**Comments to the Author**

1. Does the manuscript provide a valid rationale for the proposed study, with clearly identified and justified research questions?

Reviewer #1: Yes

Reviewer #2: Yes

2. Is the protocol technically sound and planned in a manner that will lead to a meaningful outcome and allow testing the stated hypotheses?

Reviewer #1: Yes

Reviewer #2: Yes

3. Is the methodology feasible and described in sufficient detail to allow the work to be replicable?

Reviewer #1: No

Reviewer #2: Yes

4. Have the authors described where all data underlying the findings will be made available when the study is complete?

Reviewer #1: Yes

Reviewer #2: Yes

5. Is the manuscript presented in an intelligible fashion and written in standard English?

Reviewer #1: Yes

Reviewer #2: Yes

You may also provide optional suggestions and comments to authors that they might find helpful in planning their study.

Reviewer #1: The sampling process needs to be elaborated. Selection criteria for Phase I study is clear but it should also include how the participants will be selected for phase II.

How will the researcher select the minimum of 25 participants from 53 WHO regions? What are the criteria for selection?

Is there any incentive for the participants? If yes, please mention.

Please in include the guideline for the interview, FGD.

Reviewer #2: I have reviewed the authors' responses to the reviewers' comments and the accompanying revisions to the manuscript. I find their responses to be thorough and the revisions have successfully addressed all concerns raised. All the best.

**Do you want your identity to be public for this peer review?** For information about this choice, including consent withdrawal, please see our Privacy Policy

Reviewer #1: **Yes: ** Subina Bajracharya

Associate Professor

School of Nursing

Chitwan Medical College, Nepal

Reviewer #2: **Yes: ** Nada Alaidarous

---

## [Author Response · Author response to Decision Letter 2]

4 Sep 2025

Reviewers comments

1. Does the manuscript provide a valid rationale for the proposed study, with clearly identified and justified research questions?

Reviewer #1: Yes

Reviewer #2: Yes

No action required

2. Is the protocol technically sound and planned in a manner that will lead to a meaningful outcome and allow testing the stated hypotheses?

Reviewer #1: Yes

Reviewer #2: Yes

No action required

3. Is the methodology feasible and described in sufficient detail to allow the work to be replicable?

Reviewer #1: No

Reviewer #2: Yes

We trust that the additional changes to the sampling section will address Reviewer 2’s view on methodology. Many thanks

4. Have the authors described where all data underlying the findings will be made available when the study is complete?

Reviewer #1: Yes

Reviewer #2: Yes

No action required

5. Is the manuscript presented in an intelligible fashion and written in standard English?

Reviewer #1: Yes

Reviewer #2: Yes

No action required

6. Review Comments to the Author

Reviewer #1: The sampling process needs to be elaborated. Selection criteria for Phase I study is clear but it should also include how the participants will be selected for phase II.

How will the researcher select the minimum of 25 participants from 53 WHO regions? What are the criteria for selection?

Is there any incentive for the participants? If yes, please mention.

Please in include the guideline for the interview, FGD.

Reviewer #2: I have reviewed the authors' responses to the reviewers' comments and the accompanying revisions to the manuscript. I find their responses to be thorough and the revisions have successfully addressed all concerns raised. All the best.

Thank you for this observation. we have revised the sampling section on page 4 to further strengthen and clarify our approach.

With regards to the focus group discussion/interview guide which will be used in Phase 2. This can not be included as it will be developed in light of the responses obtained in phase 1 of the study. This is stipulated on page 6, under the section heading ‘Phase 2 Online focus group. When the study is completed/reported access to the discussion/interview guide used for the study will be provided.

Many thanks.

---

## [Editor Report · Decision Letter 2]

7 Sep 2025

Digital nursing and midwifery leadership:  Protocol for a multi-method exploration of policy implementation and impact on practice in the WHO European Region

PONE-D-25-02701R2

Dear Dr. Reid,

We’re pleased to inform you that your manuscript has been judged scientifically suitable for publication and will be formally accepted for publication once it meets all outstanding technical requirements.

Kind regards,

Oluchukwu Loveth Obiora, PhD

Academic Editor

PLOS ONE

---

## [Editor Report · Acceptance letter]

PONE-D-25-02701R2

PLOS ONE

Dear Dr. Reid,

I'm pleased to inform you that your manuscript has been deemed suitable for publication in PLOS ONE. Congratulations! Your manuscript is now being handed over to our production team.

Kind regards,

on behalf of

Dr. Oluchukwu Loveth Obiora

Academic Editor

PLOS ONE